# Proteome-scale prediction of molecular mechanisms underlying dominant genetic diseases

**Mihaly Badonyi**[ID]*, **Joseph A. Marsh**[ID]

MRC Human Genetics Unit, Institute of Genetics and Cancer, University of Edinburgh, Edinburgh, United Kingdom

\* mihaly.badonyi@ed.ac.uk

**Data Availability Statement:** All data and code to build the models and reproduce the analyses have been deposited on the Open Science Framework at https://osf.io/z4dcp/.

## Abstract

Many dominant genetic disorders result from protein-altering mutations, acting primarily through dominant-negative (DN), gain-of-function (GOF), and loss-of-function (LOF) mechanisms. Deciphering the mechanisms by which dominant diseases exert their effects is often experimentally challenging and resource intensive, but is essential for developing appropriate therapeutic approaches. Diseases that arise via a LOF mechanism are more amenable to be treated by conventional gene therapy, whereas DN and GOF mechanisms may require gene editing or targeting by small molecules. Moreover, pathogenic missense mutations that act via DN and GOF mechanisms are more difficult to identify than those that act via LOF using nearly all currently available variant effect predictors. Here, we introduce a tripartite statistical model made up of support vector machine binary classifiers trained to predict whether human protein coding genes are likely to be associated with DN, GOF, or LOF molecular disease mechanisms. We test the utility of the predictions by examining biologically and clinically meaningful properties known to be associated with the mechanisms. Our results strongly support that the models are able to generalise on unseen data and offer insight into the functional attributes of proteins associated with different mechanisms. We hope that our predictions will serve as a springboard for researchers studying novel variants and those of uncertain clinical significance, guiding variant interpretation strategies and experimental characterisation. Predictions for the human UniProt reference proteome are available at https://osf.io/z4dcp/.

## Introduction

Genetic diseases are often caused by protein-altering mutations that act via different protein-level mechanisms. One such mechanism is loss of function (LOF), wherein the mutation completely ablates the function of the protein. While LOF-associated disease is often due to protein null mutations, in which little or no protein is produced due to premature termination codons and nonsense mediated decay, it can also be caused by protein-altering mutations, e.g., missense mutations that destabilise protein structure. Nearly all recessive disorders are

**Funding:** European Research Council (ERC) under the European Union's Horizon 2020 research and innovation programme (grant agreement No. 101001169) Biotechnology and Biological Sciences Research Council EASTBIO Doctoral Training Programme (BB/M010996/1) The funders had no role in study design, data collection and analysis, decision to publish, or preparation of the manuscript.

**Competing interests:** The authors have declared that no competing interests exist.

associated with LOF, with only a few rare cases of alternative mechanisms identified to date [1]. By contrast, although LOF mutations often cause dominant disease via haploinsufficiency when the lost function cannot be compensated by the remaining wild-type allele [2], many dominant disorders are caused by non-LOF mechanisms. These include gain of function (GOF), which is characterized by an altered or newly appeared function in the mutant protein, and the dominant-negative (DN) effect, whereby the mutant allele can directly or indirectly disrupt the function of the wild type [1]. Understanding the molecular mechanisms through which protein mutations exert their effects has great potential for improving diagnoses and developing adequate treatments of genetic disorders. Recent years have seen major improvements in experimental methods capable of probing a large number of variants at once. In particular, deep mutational scanning and its modifications hold huge promise for deciphering the effect of variants and, by extension, their mechanisms [3]. For now, relative to LOF, our understanding of the properties of variants that act via DN or GOF (collectively, non-LOF) mechanisms remains limited.

In many cases, the different dominant molecular mechanisms are not mutually exclusive in a gene. For example, a type of cardiomyopathy can be influenced by both titin haploinsufficiency (LOF) and truncated titin peptides that seem to "poison" wild-type complexes (DN) [4]. Although molecular mechanisms are indisputably variant-level phenomena, it has become evident that mutations in many genes are more likely to exhibit a specific mechanism [5]. This concept has shed light on certain structural and functional properties associated with non-LOF proteins at the systems level [5, 6]. Moreover, these analyses have called attention to a limitation of current variant effect predictors (VEPs), as they struggle to accurately predict the pathogenicity of non-LOF variants [5, 7, 8]. This is a critical problem to address, as otherwise there is a strong possibility that we could miss variants on the account of difficulties in computationally predicting their effects. A gene-level predictor of molecular mechanisms could potentially be very useful for highlighting genes where VEPs could perform poorly and miss pathogenic variants. Given the limitations of current methods that primarily focus on LOF mechanisms, such as haploinsufficiency predictors [9–11], we aimed to develop a method for more directly differentiating between dominant molecular mechanisms.

Previously, we built a simple model to predict genes most likely to be associated with non-LOF mechanisms [6]. Predictions from this model were incomplete for the proteome due to limited availability of protein structural and functional features, in particular its reliance on quaternary structure information that comes from experimentally determined structures. Additionally, DN and GOF mechanisms were combined into a single non-LOF category because the genes associated with them tend to have similar attributes such as evolutionary constraints, functional niches, and protein complex topologies. To address these problems, here we train three support vector machine binary classifiers using surrogate structural and functional features complete for the proteome. Each classifier is tasked with identifying one of the molecular mechanisms over another unique class (DN *vs* LOF and GOF *vs* LOF) or pooled classes (LOF *vs* non-LOF). This design maximizes the number of cases available for training and enables a flexible classification regime. Analysis of an unbiased set of disease-linked proteins with predicted mechanisms suggests that their properties, independent from model features, are consistent with known characteristics of the mechanisms, thus strongly supporting the utility of this approach.

To facilitate a community-wide effort in piecing together the human variant effect map, we make predictions for all human protein-coding genes. These predictions can pinpoint the likely mechanism of a variant in a dominant gene with no previous association to a molecular mechanism and offer valuable insight when assessing novel variants in a gene with no prior disease linkage. Additionally, they could help prioritise genes in clinical and population

genetics data for laboratory studies and allow researchers to explore the functional, structural, and evolutionary properties inherent to the mechanisms.

## Methods

### Workflow

Our strategy to predict genes associated with different dominant molecular mechanisms is illustrated in Fig 1 and detailed in the following sections.

### Software and databases

In this study, we rely on a number of properties derived from the AlphaFold-predicted structures of human proteins [12]. Solvent accessible surface area is calculated at residue level with AREAIMOL from the CCP4 programme suite [13]. FoldX version 5.0 [14] is run to determine the predicted Gibbs free energy of folding (ΔΔG) of missense substitutions by first calling the RepairPDB command followed by BuildModel. Residue-level SCRIBER scores [15] for the human proteome are extracted from the 9606_database.json file provided by the DescribePROT database [16]. Protein-level UniProt ProtNLM embeddings [17] are obtained via the UniProt FTP for the human proteome (2023_02 release). All 1024 embedding dimensions are extracted from the per-protein.h5 file using the h5dump command from the HDF5 command line tool. Clustal Omega version 1.2.4 [18] is used to create a distance matrix of protein sequences for homology filtering. Scores from VEPs ESM-1v [19], EVE [20], MetaRNN [21], and VARITY_R [22] are gathered using an in-house pipeline described in [23]. Machine learning methods are implemented in R version 4.3.0 [24] using the tidymodels framework with extension packages themis, probably, and DALEX. Statistical tests are performed with the rstatix package. We consider an alpha level of 0.05 to be statistically significant. Bootstrap confidence intervals are obtained from 1,000 resamples with the percentile method [25].

### ClinVar and gnomAD mapping to UniProt reference proteome

Genomic coordinates of pathogenic and likely pathogenic (hereinafter "pathogenic") missense variants are extracted from the ClinVar [26] variant calling file (VCF) (accessed on 2023-06-15) with BCFtools [27]. Putatively benign variants (hereinafter "benign") with a "PASS" filter are extracted from gnomAD v2.1.1 [28] using the exomes all chromosomes VCF file. Mapping

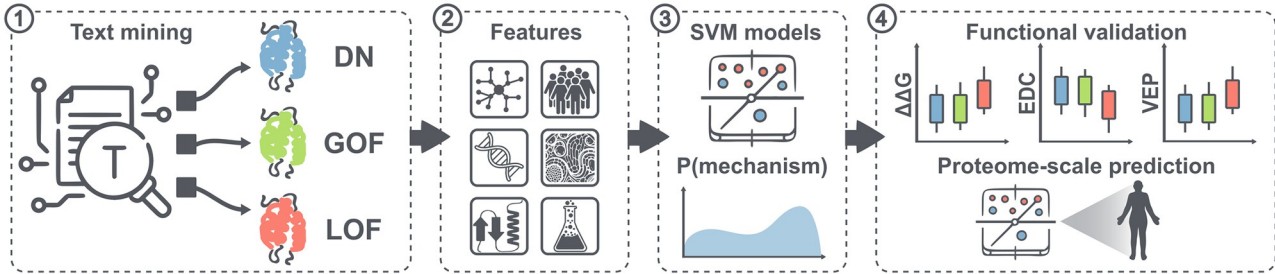

**Fig 1. Data acquisition and model building workflow. 1)** Identification of reported dominant molecular mechanisms in the literature and clinical databases. **2)** Collection of diverse gene-level features that are known to discriminate between molecular mechanisms. **3)** Training of 3 semi-independent support vector machine classifiers to predict genes most likely to be associated with DN, GOF and LOF molecular mechanisms. **4)** Validation of predictions via conventional cross-validation, as well as through the examination of functional and structural properties of an independent test set.

is performed with Ensembl VEP 107 [29] using the—uniprot and—canonical flags. For each variant, we select either the canonical transcript or the first UniProt isoform on condition that the mapped amino acid is identical to that in the sequence.

## Data acquisition

Evidence for DN, GOF, or LOF molecular disease mechanism observed in a gene was collected using a combination of text-mining and semi-manual curation, as previously described [6]. We updated the gene set using newly added autosomal dominant genes from the Online Inheritance of Man (OMIM) database [30] (accessed via the API on 2023-05-25) and developmental disorder genes with monoallelic requirement from the Deciphering Developmental Disorders Genotype-to-Phenotype (DDG2P) database (https://www.deciphergenomics.org/ddd/ddgenes/, accessed on 2023-05-25). The updated list contains 1,270 genes with one or more molecular mechanism class based upon available evidence. Of the genes, 874 are assigned to a unique class, 318 are assigned to two classes, and 78 genes intersect with all three classes. We provide the set of manually curated genes, their class labels and evidence lines along with their PubMed identifiers in S1 Table.

## Feature selection and engineering

We use gene and protein-level features previously described in [6], notably excluding subunit interface size, which is derived from experimentally determined structures of protein complexes, and the PANTHER functional classification of proteins [31], which is incomplete for the human proteome. We sought to rationally select and design surrogate features that make up for the loss of protein structural and functional information. To replace interface size, we calculate the median SCRIBER score of residues that have >5% relative surface accessible surface area (RSA) in the corresponding AlphaFold predicted structure. RSA is defined as the ratio of the maximum solvent accessible surface area determined from Gly-X-Gly tripeptides (where X denotes the amino acid) [32] and that of the residue in the context of the monomeric protein. To substitute protein functional classification, we use UniProt ProtNLM embeddings that capture information about Pfam domains and functional sequence features and transfer those to unannotated and manually unreviewed entries. In each model, to keep the ratio of primary cases to number of predictors high and avoid overfitting on a large number of uncorrelated features, we select the top 20 ProtNLM embeddings by Wilcoxon effect size between the relevant binary outcomes. In addition, we introduce a pair of features related to the likelihood of randomly drawing a missense substitution with the characteristics of a particular type of molecular mechanism. This heuristic metric is derived as the median of the ratio of ESM-1v score and the FoldX-computed $\Delta\Delta G$ of all missense mutations based on possible amino acid substitutions in the canonical sequence. We use both raw $\Delta\Delta G$ and its absolute value ($|\Delta\Delta G|$), which has been shown to improve the performance of pathogenic variant prediction [33]. Finally, we complement existing population genetics constraint metrics with $s_{\text{het}}$ [34], which relates the frequency of protein null mutations in a gene to the strength of selection against them, with higher values indicating stronger selection and lower tolerance to LOF mutations. We provide the list of all features used in the models and their detailed descriptions in S2 Table.

## Model design and data preprocessing

We opted for three separate binary classifiers rather than one multiclass predictor for several reasons. First, a multiclass approach would require seven classes to account for all mixed class combinations. The case numbers of the mixed classes are not sufficient for modelling,

as they prohibit cross-validation and necessitate the use of fewer features to avoid overfitting [35]. Second, the prevalence of different mechanisms within the same gene is often unknown, making it difficult to weight them appropriately in a multiclass setting. Since many genes with a prevailing mechanism can have rare occurrences of other mechanisms, our current mixed-class groups are likely subject to a level of study bias, which would be unfavourably reflected in the model output. Third, binary models allow genes with mixed classes to contribute to more than one model, delegating the task of weighting mechanism propensities to the models. Thus, multiple binary models can essentially function like a multiclass predictor, without the data scarcity problems introduced by attempting a direct multiclass prediction [36].

In each model, we prioritise the first event level of the binary outcome. For example, in the DN *vs* LOF model, we treat any gene with a DN class annotation as DN even if it has other class assignments; this strategy is likewise applied to the GOF *vs* LOF and the LOF *vs* non-LOF models (where non-LOF represents pooled DN and GOF gene sets). We next remove features that have a Spearman correlation >0.9 with another, which eliminates the gnomAD oe_lof metric (observed over expected ratio for predicted LOF variants in transcript) in all three models due to its high correspondence with $s_{het}$. The features are normalised and missing values are imputed based upon all other features with five nearest neighbours [37]. We use a distance matrix of the protein sequences to create a non-redundant dataset with proteins sharing <50% sequence identity within each outcome, e.g., within DN or LOF genes, but not across, as it may be important to learn differences between homologues assigned to different classes. To improve the signal-to-noise ratio, when possible, instead of randomly removing one protein from a pair of proteins above the sequence identity cutoff, we remove the one that overlaps with another class. This procedure results in the following class proportions across the models (protein count, class percentage): DN (498, 52%) *vs* LOF (451, 48%), GOF (479, 51%) *vs* LOF (463, 49%) and LOF (610, 52%) *vs* non-LOF (559, 48%).

## Initial model screen

To assess which statistical learning approach suits the data and the classification problem the best, we carried out a model screen. We chose five diverse modelling methods (tuned hyperparameters): lasso logistic regression (penalty), multilayer perceptron (penalty, hidden units, epochs), polynomial support vector machine (SVM) (cost, degree, scale factor, margin), and two tree-based methods, random forest (trees, mtry [minimum number of randomly sampled features at each split], min_n [minimum number of data points a node must contain]) and the LightGBM algorithm (trees, tree depth, mtry, min_n, loss reduction) [38]. Maintaining class ratios, we split the data into 75% training and 25% test sets and ran a 10-fold cross-validation, similarly divided into 75% analysis and 25% assessment sets. An efficient grid search was performed via the ANOVA race tuning method [39] on a random grid of size $k = 10\times$(number of hyperparameters), which eliminated configurations unlikely to have been the best after 3 resamples using repeated measures ANOVA. We compared the models by both the area under the receiver operating characteristics (AUROC) curve of the ANOVA race-winning configuration measured on the cross-validation folds as well as the AUROC of the $k$ random grid configurations of each model on the test set (Fig 2). By the former, the top-ranking models were LightGBM for DN *vs* LOF and LOF *vs* non-LOF models and polynomial SVM for the GOF *vs* LOF model. However, because the $k$ random grid configurations of the polynomial SVM consistently gave the best performance on the test sets, suggesting it performs well on unseen data without hyperparameter tuning, it was chosen to reduce the risk of overfitting and to maintain a similar prediction profile across the three models.

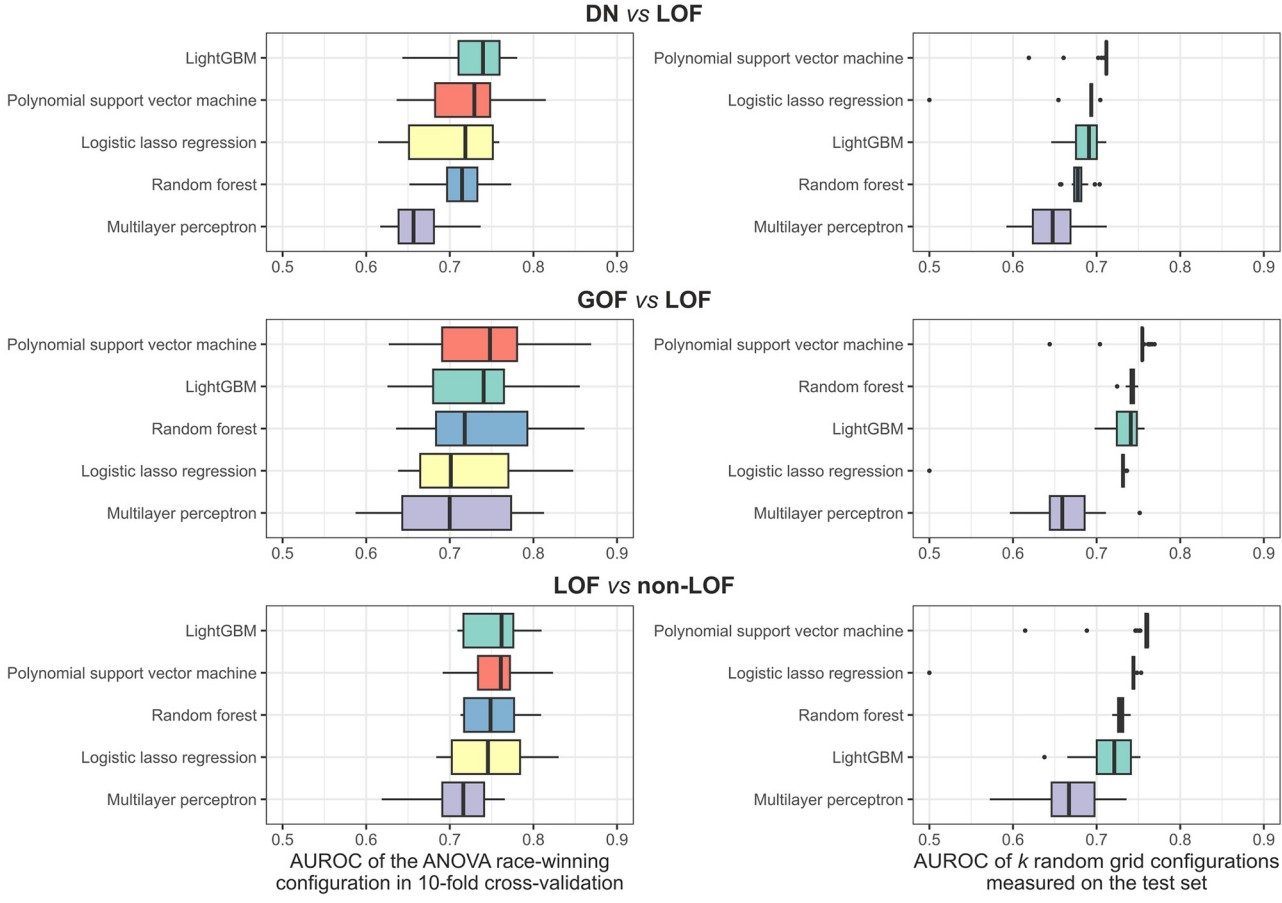

**Fig 2. Results of the initial model screen.** Plots on the left show the AUROC of the best hyperparameter set of each model on the cross-validation folds. Plots on the right show the AUROC of *k* randomly generated parameter combinations as measured on the test sets, where *k* = 10×(number of hyperparameters). Boxes denote data within 25th and 75th percentiles and the middle line represents the median. Whiskers extend from the box to 1.5× the interquartile range.

## Model building

The final DN *vs* LOF, GOF *vs* LOF, and LOF *vs* non-LOF models are built using an SVM with a polynomial kernel. The data are divided into 75% training and 25% test sets with 3-times repeated 10-fold cross-validation, which itself is split into 75% analysis and 25% assessment sets. We perform Bayesian hyperparameter optimisation on the tunable parameters of the model: cost, degree, scale factor, and margin. Initially, a random grid of size 10 is created for the Gaussian process model and a maximum of 20 search iterations are performed, with an early stop after 10 cycles if no improvement is observed. The models are finalised based upon the parameters that achieve the highest AUROC. Exact values of the SVM hyperparameters used in the models are provided in S2 Table.

## Global and local model feature importance assessment

Global feature importance is estimated by loss in AUROC upon removal of the feature in 10 permutations. Local influence of features on the predicted outcome for a protein is estimated via computing the features' Shapley values [40] in 20 permutations.

### Model evaluation

We compute threshold-agnostic performance measures for the models based upon the test sets, including AUROC and area under the precision recall curves (AUPRC) (Fig 3). The consistency of AUROCs measured on the test sets relative to the cross-validation folds suggest the models are not overfitted (DN *vs* LOF: $AUROC_{CV}$ = 0.718, $AUROC_{test}$ = 0.71; GOF *vs* LOF: $AUROC_{CV}$ = 0.73, $AUROC_{test}$ = 0.763; LOF *vs* non-LOF: $AUROC_{CV}$ = 0.739, $AUROC_{test}$ = 0.763). To assess the models in further analyses, we determine the probability thresholds at which the models reach 50% sensitivity on their respective test sets. At these thresholds, which we refer to as $t_{50}$, the models are able to correctly identify half of the positive cases and approximately 80% of the negative cases. Threshold-dependent performance measures at $t_{50}$, including accuracy, Matthews correlation coefficient (MCC) and the F1 score, are summarised in Table 1.

Biologically interpretable evaluation is performed by examining the models' capacity to identify emergent properties known to be associated with dominant molecular disease mechanisms [5, 6]. Specifically, we evaluate 3 properties:

1. The energetic impact of pathogenic missense mutations in the structure, approximated by the FoldX-predicted Gibbs free energy of folding (ΔΔG). Missense mutations that map to residues with a predicted local distance difference test (pLDDT) value [41] less than 70 are excluded. This is because residues with low pLDDT are less likely to be ordered and thus ΔΔG is less interpretable for them.

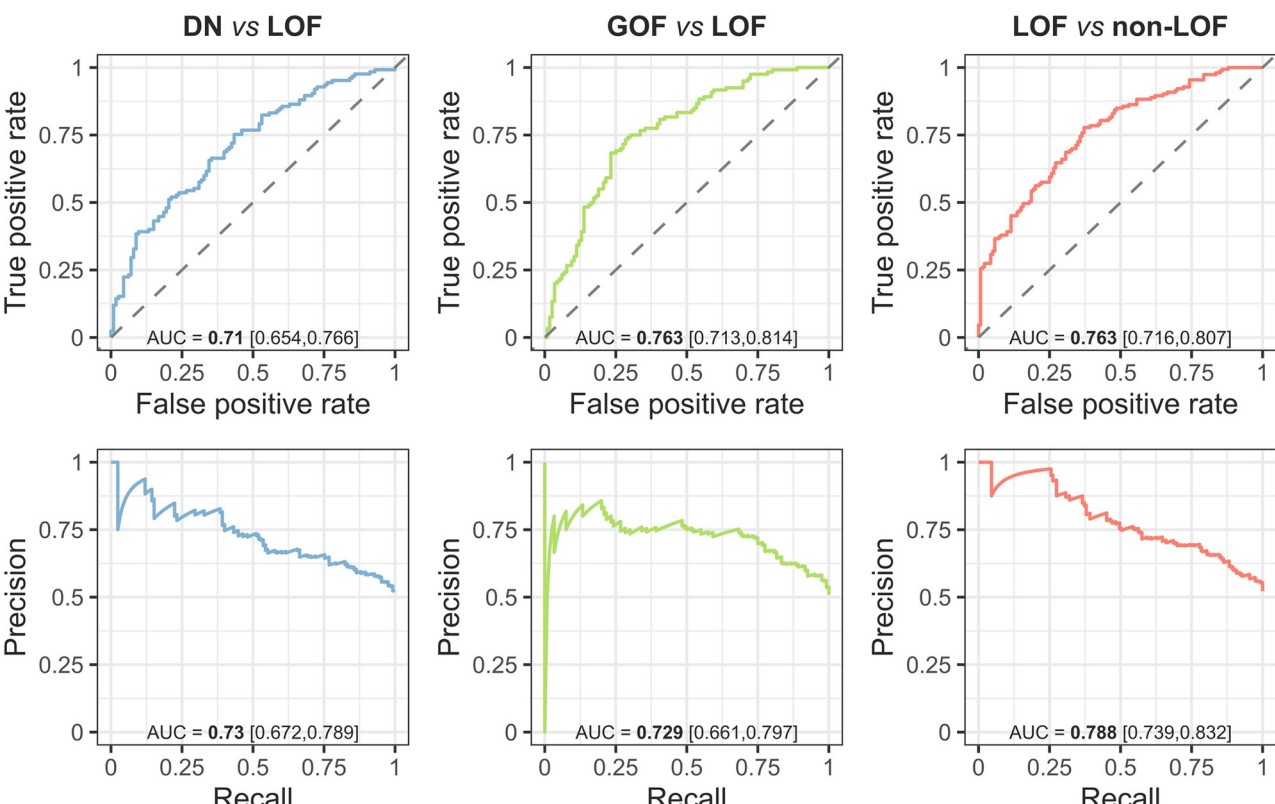

**Fig 3. ROC and PR curves of the models measured on the test sets.** Numbers in bold denote point estimates and those in brackets represent 90% bootstrap confidence intervals.

**Table 1.**

| Model | $t_{50}$ | Sensitivity | Specificity | Accuracy | MCC | F1 |
|---|---|---|---|---|---|---|
| DN *vs* LOF | 0.61 | **0.519** [0.448,0.584] | **0.778** [0.717,0.841] | **0.642** [0.592,0.693] | **0.306** [0.208,0.408] | **0.603** [0.539,0.664] |
| GOF *vs* LOF | 0.63 | **0.519** [0.45,0.592] | **0.818** [0.759,0.879] | **0.666** [0.619,0.712] | **0.353** [0.258,0.449] | **0.612** [0.547,0.676] |
| LOF *vs* non-LOF | 0.64 | **0.519** [0.451,0.588] | **0.815** [0.764,0.871] | **0.66** [0.618,0.703] | **0.348** [0.263,0.435] | **0.614** [0.556,0.672] |

Threshold-dependent performance metrics.

Metrics derived for the probability thresholds at which the models have approximately 50% specificity on the test sets ($t_{50}$). Numbers in bold denote point estimates and those in brackets are 90% bootstrap confidence intervals.

2. The degree to which pathogenic missense mutations spatially cluster in the structure, approximated by the extent of disease clustering (EDC) metric. EDC is calculated as previously described [5], however, alpha carbon atoms with a pLDDT<70 are excluded from the calculation and only proteins with at least 5 pathogenic variants after this procedure are used for the analysis. This is because pathogenic missense mutations are enriched in structured regions of proteins [42], hence AlphaFold structures with high disorder could exhibit pseudo-clustering.

3. The performance of pathogenic missense variant prediction in a binary classification task, using two unsupervised (ESM1-v and EVE) and two supervised (MetaRNN and VARI-TY_R) VEPs, measured by AUROC. Only proteins with at least 5 pathogenic and 5 benign variants, and all variants mutually shared across the VEPs (that is, have predicted values), are considered for the analysis to increase per-protein AUROC confidence.

## Statistical overrepresentation test for molecular function

Functional enrichment of proteins that are subsets of predicted DN or GOF proteins is performed by comparing the protein lists against each other via PANTHER [43] using Fisher's exact test and the Gene Ontology release 2023-05-10. We consider functions with a false discovery rate <1%, a sample size (number of proteins) >50, and a fold-enrichment of >1.5. To avoid redundancy, we select a single term with the largest sample size for groups of similar terms.

## Results and discussion

### Global and local feature importance evaluation

In total, 21 interpretable features were used in the models, including properties derived from protein sequences, structures, networks, and gene mutational constraints [6]. Additionally, 20 language model-based embeddings were also included, which are thought to represent protein function in their latent space [44]. As a measure of feature importance, we calculated the loss in AUROC relative to the full model (Fig 4). While we cannot draw conclusions on why a particular ProtNLM embedding is useful for a given model, their relatively high ranks in the GOF *vs* LOF model may reflect that those genes whose dominant disease mutations act via GOF are functionally more dissimilar or diverse than DN or LOF genes. In terms of the interpretable features, $s_{het}$, a metric that relates the frequency of LOF mutations in a gene to the strength of selection against them, ranks 1st and 3rd in the DN *vs* LOF and LOF *vs* non-LOF models, respectively. This emphasizes the importance of eliminating length-bias from selective constraint metrics to improve their power [34]. Moreover, the number of paralogues a protein has

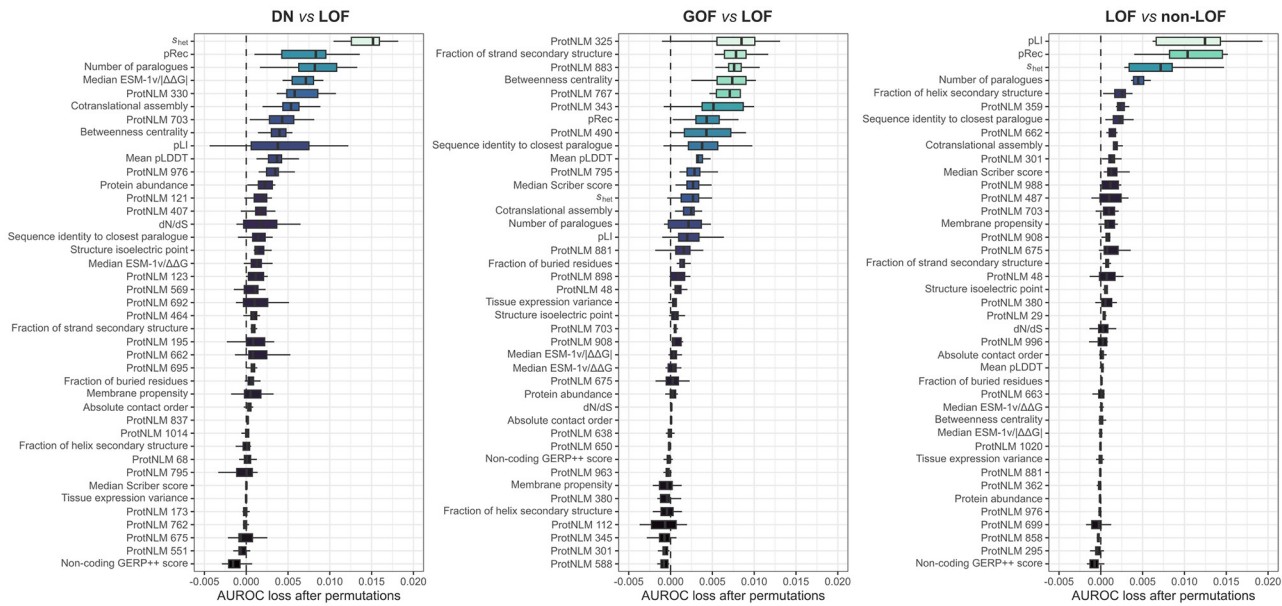

**Fig 4. Feature importance of the models.** Feature importance is estimated by loss in AUROC upon removal of the feature from the model in 10 permutations. Boxes denote data within 25th and 75th percentiles and the middle line represents the median. Whiskers extend from the box to 1.5× the interquartile range. Dashed line is at AUROC loss = 0.

in the genome seems similarly important for the latter two models, ranking 3rd and 4th, respectively, consistent with our previous results [6]. Other notable features are gnomAD metrics pLI, the probability of the gene being intolerant to loss of function, and pRec, the probability of the gene being associated with recessive disease, which rank consistently high. This is unsurprising given that genes with high pLI should be enriched in genuinely haploinsufficient genes and those with non-LOF disease mechanisms will tend to have higher pRec values, i.e. are more "recessive-like". Interestingly, the median ESM-1v/|ΔΔG| metric ranks 4th in the DN *vs* LOF model, suggesting that DN genes may possess a missense variant repertoire that is biased for functionally important but structurally less damaging effects.

We next looked at the worst and best predicted genes from the training sets of each model and calculated Shapley values for the features. These values indicate the average contributions of different feature orderings, with positive values suggesting a tendency to predict a protein towards the primary outcome and vice versa. In the training set of the DN *vs* LOF model, the DNA-binding protein SATB2 has the lowest probability of belonging to the DN class (pDN), while the type I cytoskeletal keratin 14 (KRT14) has the highest (Fig 5). Shapely values for SATB2 suggest that primarily $s_{het}$, but also, for example, pLI and cotranslational assembly have contributed to its low pDN. Indeed, SATB2 has a particularly high $s_{het}$ (0.652, which is 5.2 standard deviations above the proteome mean), a pLI of 1, and has been found to cotranslationally assemble in a recent study [45], consistent with a decreased likelihood of observing its role in disease through a DN mechanism [6]. In our source data, evidence for SATB2 dominant negativity came from a report that describes a frameshift mutation in the gene likely to escape nonsense-mediated decay, whose clinical phenotype is more severe than glass syndrome, attributed to SATB2 haploinsufficiency [46]. However, SATB2 is recognised by the ClinGen review panel to have sufficient evidence for haploinsufficiency [47]. Thus, it is possible that SATB2 either represents a false positive case or, due to regression to the mean, the model fails to identify it as its features are too consistent with the secondary outcome. In

contrast, Shapely values for KRT14 indicate congruence with known characteristics of DN proteins, e.g., the high number of paralogues (68 in the proteome, only 1 in the training set). Furthermore, keratin disorders are considered a classical group of DN diseases [48], reinforcing the prediction of the model.

Similar conclusions can be drawn for the other two models. AF4/FMR2 family member 4 (AFF4), implicated in CHOPS syndrome, is a GOF case in the source data with the lowest pGOF value among genes of the training set (Fig 5). Three heterozygous missense mutations in its gene were recently found to cause GOF due to decreased clearance of the protein by the ubiquitin proteasomal system, leading to transcriptional overactivation [49]. Since transcription factors are frequently haploinsufficient [50], their gene-level features in our data may generally align more with a LOF mechanism, as supported by the Shapley values of pLI and $s_{het}$ for AFF4, which could hinder the identification of similar edge-cases. Interestingly, the worst predicted positive case by the LOF vs non-LOF model is apolipoprotein E (APOE) (Fig 5), which does lack sufficient evidence for dosage sensitivity in ClinGen. On further review, evidence in

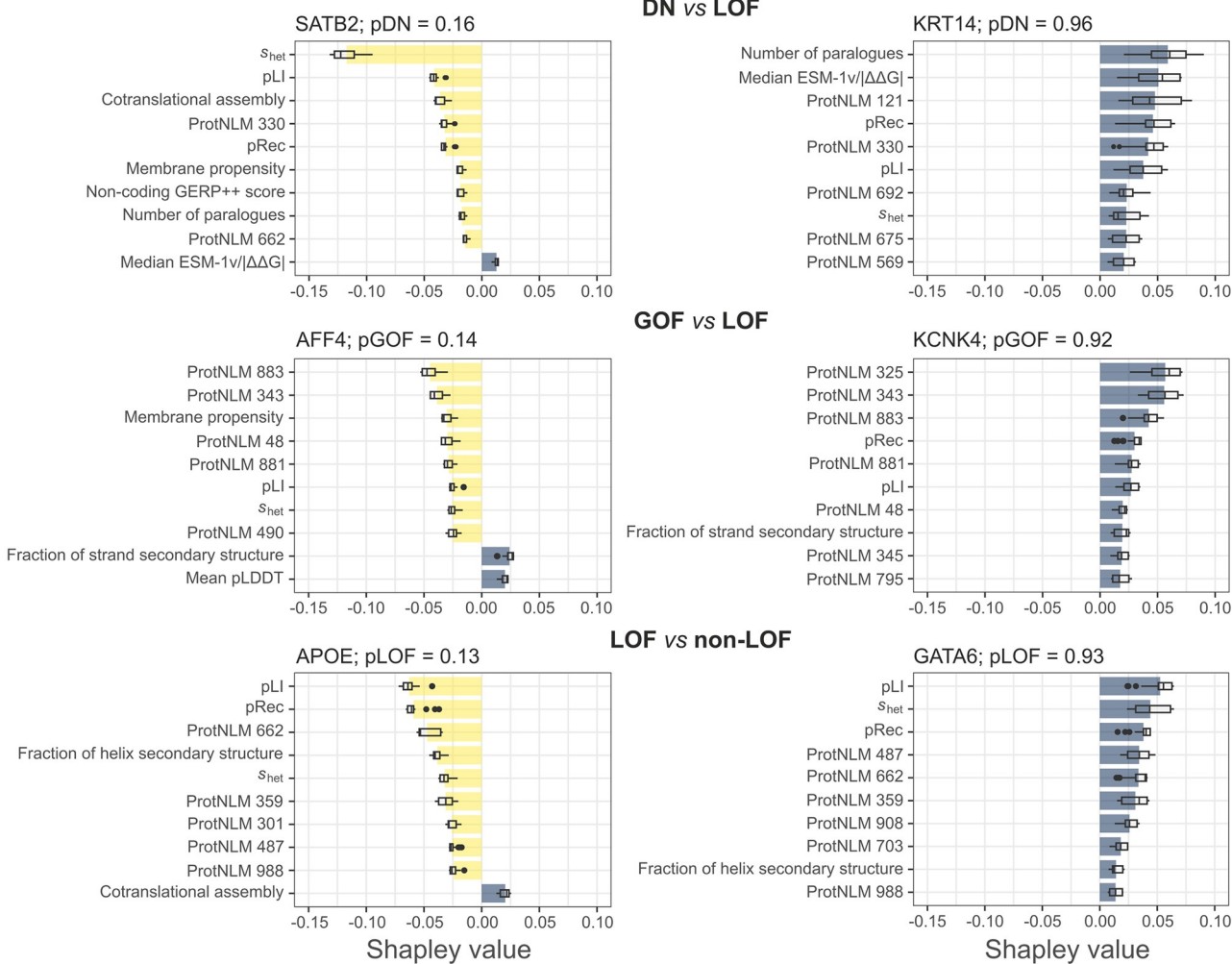

**Fig 5. Local interpretations of model class probabilities.** Plots show the worst (left) and best (right) predicted genes of the training sets and Shapley values of the top 10 most influential features in 20 permutations. Bars show the mean, boxes denote data within 25th and 75th percentiles and the middle line is the median. Whiskers extend from the box to 1.5x the interquartile range.

our source data revealed that a heterozygous null mutation of APOE had been linked to protection against a type of amyloidosis observed in Alzheimer's disease, rather than being implicated in disease causation [51]. Hence, the model is able to correctly down-weight false positive cases.

## Proteome-scale molecular mechanism prediction

From the test sets we derived probability threshold $t_{50}$ for each model, which represents a classification performance where approximately half of the cases belonging to the primary outcome and 80% of the secondary outcome are correctly predicted (Table 1). This threshold constitutes a relatively good balance between sensitivity and specificity while being more stringent than what is commonly considered the optimal threshold, e.g., minimum distance from the [0,1] corner of the ROC curve (dashed vertical line in Fig 6A), and being more lenient than the maximum positive predictive value. To allow the specification of different thresholds, we provide estimates of common threshold metrics in S2 Table for every 0.01 increase in class

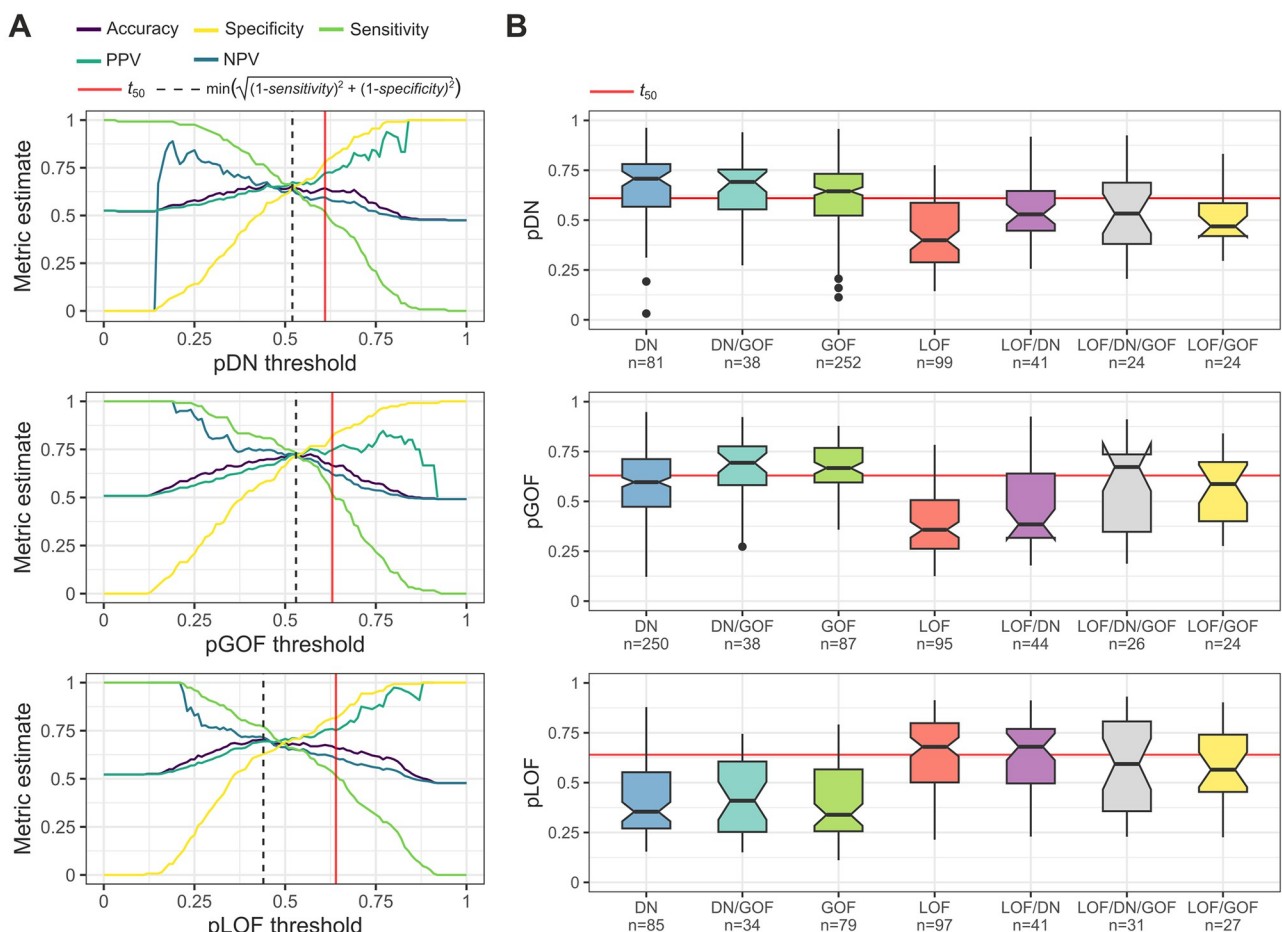

**Fig 6. Threshold plots and test set class probabilities.** (**A**) Commonly used threshold metrics derived from the test sets for pDN, pGOF, and pLOF. NPV and PPV denote negative and positive predictive values, respectively. Estimates for these metrics are provided in S2 Table for every 0.01 increase in class probability. (**B**) Model probabilities mapped to the test sets, classified by their ground truth classes, both unique and overlapping. Sample sizes indicate the number of proteins. Boxes denote data within 25th and 75th percentiles and the middle line represents the median. Whiskers extend from the box to 1.5x the interquartile range and the notch contains the 95% confidence interval of the median. Dashed lines run at the $t_{50}$ probability thresholds.

probability. In Fig 6B, we show $t_{50}$ in the context of model probability distributions for unique and overlapping cases in the test sets. Intriguingly, both the DN *vs* LOF (pDN) and the GOF *vs* LOF (pGOF) models assign higher probabilities to mixed LOF/GOF and LOF/DN classes, respectively, relative to the unique secondary outcomes, despite the fact that the models are blind to the overlapping classification of these cases. This suggests that these genes may have intermediary characteristics or exhibit a higher false positive rate for the secondary class in the ground truth data.

We chose a distribution-agnostic approach to classify the human proteome into a single molecular mechanism class. By ranking the class probabilities rather than comparing their raw values, the approach is less sensitive to differences in the properties of the models' class probability distributions. First, class probabilities are computed and ranked for all proteins and then each protein is assigned to the mechanism with the highest rank relative to the proteome, on condition that its class probability is above the $t_{50}$ value. Based upon this strategy, in the 2023_02 UniProt reference proteome, 6,058 proteins are predicted to be DN, 5,287 GOF, and 2,580 LOF, with 6,440 proteins lacking classification.

It is important to emphasize that the models do not predict disease involvement, i.e. the above proportions should not be misconstrued as implying a twofold prevalence of DN over LOF proteins contributing to disease. Only dominant disease genes were included in training, so the models were not designed to distinguish disease-associated from non-disease proteins. Instead, they can be interpreted as representing the most likely molecular mechanism by which mutations in a given gene could cause disease, if it was a dominant disease gene. The high number of predicted DN and GOF proteins may be ascribed to a higher number of paralogous proteins in these mechanism classes relative to LOF [6], which results in a larger fraction of the proteome aligning with their characteristics, regardless of whether or not they actually cause disease when mutated. When we consider only known dominant disease genes, the proportion of proteins predicted to be DN, GOF, and LOF are 27%, 28%, and 44%, respectively.

In practical scenarios, researchers are often concerned with anticipating a LOF or an alternative molecular mechanism when encountering missense variants in a gene. In such a situation, an increased pDN or pGOF score would suggest that the gene's characteristics deviate from those typically linked to LOF genes, a notion that would be further supported by a low pLOF score. Although the models do not excel at identifying positive cases at an apparent sensitivity level of 50%, they are much better at identifying negative cases. Therefore when the researcher is confident that the dominant missense variants are linked to a particular phenotype, the higher-ranking molecular mechanism is still more likely to be correct.

Importantly, however, relative to haploinsufficiency predictors introduced in previous studies [9–11] (comparisons with pLOF in S1 Fig), our models are unique in that they were trained on strictly autosomal dominant genes and specifically designed to tell apart alternative molecular mechanisms from that of LOF. Thus, we expect them to be much more useful than haploinsufficiency predictors for the interpretation of missense variant effects, as they more accurately represent the properties of the underlying mechanisms. We provide the class probabilities, the rank-based classification results with and without $t_{50}$ inclusion, and the labels for training set genes in S3 Table for all human proteins (n = 20,365).

## Biologically and clinically relevant validation of the models

We wished to test whether the models are able to recapitulate known properties of dominant molecular mechanisms [5]. These properties include the energetic impact of missense mutations in the protein structure, their degree of spatial clustering, and their current predictability

by state-of-the-art VEPs (see Methods). None of these properties were explicitly used for training, thus making them simple and powerful means to assess the usefulness of the predictions. We created an unbiased analysis set where we removed genes used for training and those associated with autosomal recessive inheritance. We first looked at the predicted $\Delta\Delta G$ of pathogenic missense mutations across the different predicted classes in this dataset (Fig 7A). The results suggest mutations in DN and GOF proteins are significantly less damaging than in LOF proteins. This can be explained by the idea that destabilisation is one of the signature mechanisms of LOF mutations [5]. By contrast, GOF mutations should not be too damaging in order to alter a function, and most DN and many GOF genes are in fact assembly-mediated [1]. Thus, the effect of their mutations should not preclude protein complex assembly, which makes them necessarily less damaging.

Next, we looked at how much pathogenic missense mutations cluster in the structures of proteins predicted to be DN or GOF. Our prior expectation is that both classes should exhibit higher degree of clustering than LOF, as measured by the EDC metric. We found this assumption to hold up, with both DN and GOF proteins having significantly higher clustering values than LOF proteins (Fig 7B). This observation agrees with the concept that LOF mutations are generally more sparsely distributed in the protein structure, but non-LOF mutations tend to be concentrated at protein interfaces and functional sites [5].

Lastly, we took advantage of an important bottleneck of contemporary VEPs, which is that they less well predict pathogenic missense mutations associated with DN and GOF genes [5]. In Fig 7C, we show a per-protein aggregated AUROC analysis of pathogenic *vs* benign missense variants, evaluated by two unsupervised (ESM1-v and EVE) and two supervised (MetaRNN and VARITY_R) VEPs, which we recently showed to be the top performing VEPs of their category [23]. Expectedly, missense mutations in proteins classified as LOF are much better predicted than those in DN or GOF proteins.

Importantly, the above trends are observed with the raw rank-based classification results without the $t_{50}$ probability cutoff (S2 Fig), demonstrating that they are not an artefact of a careful threshold selection. We conclude that the models effectively reproduce biologically and clinically relevant properties of dominant molecular mechanisms.

## The functional landscape of predicted DN and GOF proteins

Finally, given that pDN and pGOF class probabilities are positively correlated by design (Fig 8), we decided to explore the molecular functions associated with proteins predicted exclusively as DN or GOF using statistical overrepresentation tests. It is well-established that DN effects are common in homomers [6, 52], which are proteins that form complexes with copies of themselves. Correspondingly, we identified functions closely related to homomeric symmetry groups in exclusively DN proteins, including "organic cyclic compound binding", "oxidoreductase activity", and "hydrolase activity" [53]. Interestingly, a strongly enriched, large group of proteins possess "nucleic acid binding" function, potentially indicating a higher prevalence of DN effects among transcription factors and other DNA/RNA-binding proteins. This underscores the importance, and difficulty, of distinguishing between mutations that act through DN effects versus haploinsufficiency. In contrast, exclusively GOF proteins exhibit functions less directly associated with protein complexes but more susceptible to the impact of overactivation events. These functions include "molecular transducer activity", "signaling receptor activity", "kinase activity", "enzyme regulator activity", and "phosphotransferase activity".

Altogether, these findings reinforce that the predictions are consistent with known and expected properties of non-LOF molecular mechanisms. Thus, they can be used to test

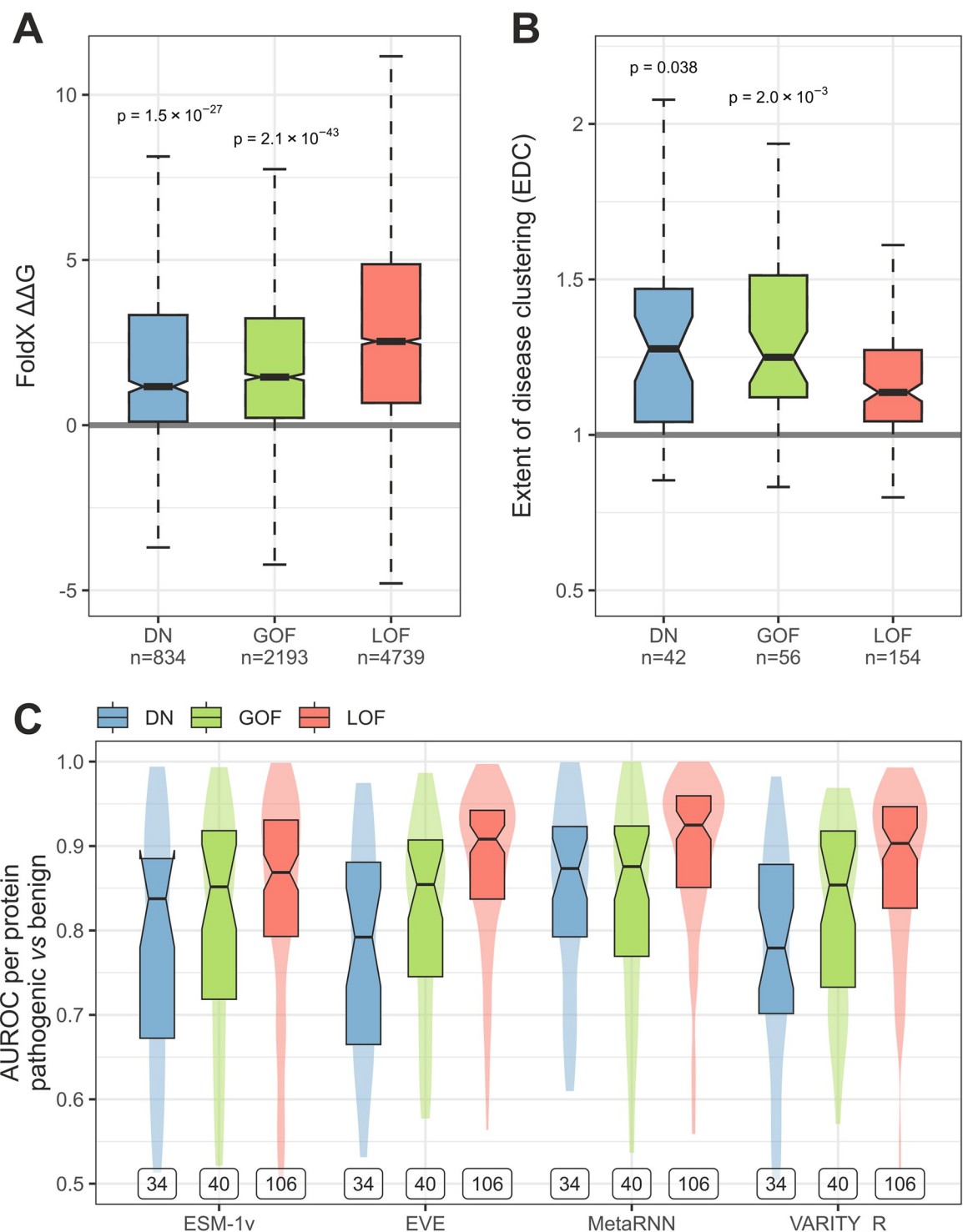

**Fig 7. Validation of the models through model-independent metrics on an unbiased analysis set.** (**A**) FoldX-predicted ΔΔG of pathogenic missense mutations. Numbers below classes denote the number of mutations. Holm-Bonferroni corrected p-values above DN and GOF boxes are relative to the LOF group and were determined by one-sided Wilcoxon rank-sum test. Sample sizes indicate the number of variants. (**B**) Class probabilities of the analysis set *vs* EDC. Sample sizes indicate the number of proteins in each class. Holm-Bonferroni corrected p-values above DN and GOF boxes are relative to the LOF group and were determined by one-sided Wilcoxon rank-sum test. (**C**) Aggregated AUROC analysis of pathogenic *vs* benign variants in predicted molecular mechanism classes. Labels indicate the number of proteins in each class. Boxes denote data within 25[th] and 75[th] percentiles, the middle line represents the median and the notch contains the 95% confidence interval of the median. Violins show area-normalized distributions.

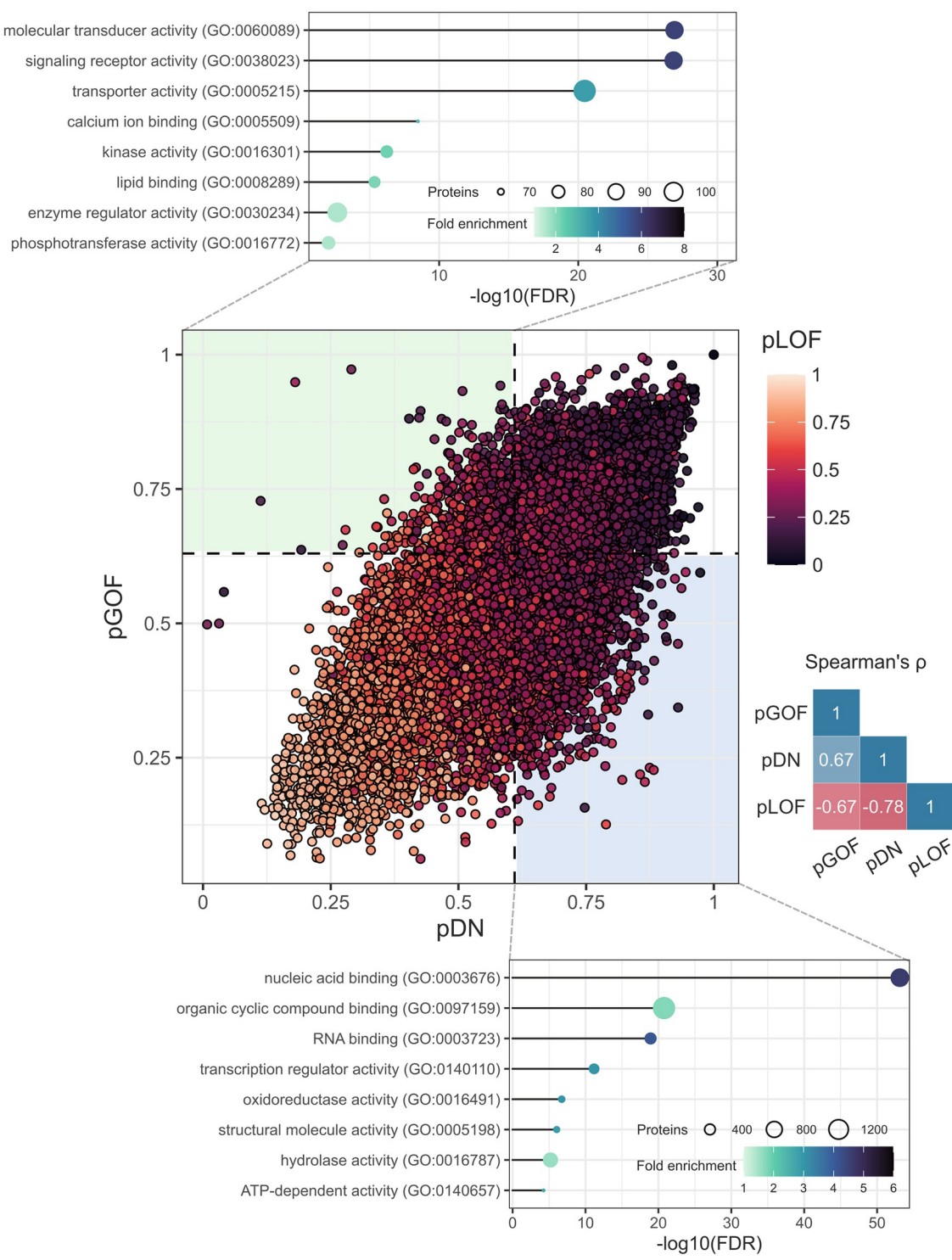

**Fig 8. The functional landscape of proteins predicted to be exclusively DN or GOF.** *Middle*: pDN *vs* pGOF scatter plot coloured by pLOF. Correlation triangle on the right shows Spearman correlations between class probabilities. *Top*: The lollipop chart shows the most enriched molecular functions for genes predicted to be GOF below the $t_{50}$ threshold for pDN (exclusively GOF). *Bottom*: Likewise, the chart shows the most enriched functions for DN genes below the $t_{50}$ threshold for pGOF (exclusively DN). The *x*-axis shows the negative $\log_{10}$ false discovery rate.

hypotheses about the functional attributes of the proteins associated with the molecular mechanisms, which could facilitate the discovery of novel features for the next generation of VEPs.

## Conclusion

We constructed three binary classifiers to predict DN, GOF, and LOF dominant molecular disease mechanisms by substituting structural and functional features of protein-coding genes with limited coverage. These models have a similar performance to our previous non-LOF *vs* LOF classifier, which was based upon such limited but interpretable features. We combined the models' output into a single class prediction using a proteome rank-based approach and examined the properties of proteins assigned to the different molecular mechanisms. Our analyses provide strong evidence that the models reproduce the properties linked to molecular mechanisms at systems level. Notably, our findings underscore the predictability of pathogenic mutations in DN and GOF proteins–a problem recognised to be more challenging than in LOF proteins. This observation was consistently upheld in our unbiased analysis set, further stressing the need for diversifying the predictive capabilities of future VEPs.

Interestingly, by examining the mechanism probabilities assigned to genes with mixed mechanism classes, we found some indication that the models have learnt the properties of genes in which multiple disease mechanisms can arise. However, our tripartite model does not directly account for different molecular mechanisms that may coexist within a gene. Therefore, considering the properties of missense variants linked to distinct diseases may improve prediction accuracy at both the phenotype and gene levels. Enhancing these predictions will be particularly beneficial for clinicians, as it will enable them to focus on variants associated with a given phenotype and obtain more precise, context-aware mechanism predictions.

We have made available the predictions for the human reference proteome, and we hope researchers will use the data to prioritise novel or challenging variants and test hypotheses about the functional roles fulfilled by these proteins. It is important to emphasize that the models do not predict the likelihood of disease involvement by a gene. Instead, a non-disease associated gene with a predicted mechanism implies that the protein's properties are most consistent with the given mechanism, which we expect will be particularly useful for researchers seeking to determine the molecular mechanisms underlying novel disease mutations.

## Supporting information

**S1 Fig. Comparison of pLOF with haploinsufficiency predictors.** (**A**) Scatter plots of haploinsufficiency scores from the indicated studies *vs* pLOF, assessed on the LOF *vs* non-LOF test set. pLOF has the highest agreement with the Shihab *et al.* predictions (Pearson's r = 0.54, p = $4.5 \times 10^{-24}$) and the lowest with those of Steinberg *et al.* (r = 0.35, p = $6.9 \times 10^{-19}$). (**B**) Pearson correlation triangle showing mutually shared values of all four metrics on human genes (n = 15,046). Although considerable variation exists even among haploinsufficiency predictors, low correlation with pLOF is expected, because haploinsufficiency predictors were not exclusively trained on dominant genes and their negative class comprised of haplo-sufficient genes of mostly no disease relevance. By contrast, pLOF values from the LOF *vs* non-LOF model were obtained by training the model to recognise haploinsufficient genes against a background of strictly dominant genes with molecular mechanisms other than simple LOF.
(TIF)

**S2 Fig. Supplemental analysis to Fig 6: Validation of the models through model-independent metrics on an unbiased analysis set, using rank-based classification without $t_{50}$ cutoff.**

(**A**) FoldX-predicted ΔΔG of pathogenic missense mutations. Numbers below classes denote the number of mutations. Holm-Bonferroni corrected p-values above DN and GOF boxes are relative to the LOF group and were determined by one-sided Wilcoxon rank-sum test. Sample sizes indicate the number of variants. (**B**) Class probabilities of the analysis set *vs* EDC. Sample sizes indicate the number of proteins in each class. Holm-Bonferroni corrected p-values above DN and GOF boxes are relative to the LOF group and were determined by one-sided Wilcoxon rank-sum test. (**C**) Aggregated AUROC analysis of pathogenic *vs* benign variants in predicted molecular mechanism classes. Labels indicate the number of proteins in each class. Boxes denote data within 25$^{th}$ and 75$^{th}$ percentiles, the middle line represents the median and the notch contains the 95% confidence interval of the median. Violins show area-normalized distributions.
(TIF)

**S1 Table. Molecular disease mechanisms collected from literature reports.** This CSV format table contains the molecular mechanisms associated with dominant genes collected from the literature. The columns are as follows: gene–gene name; class–one of "dn", "gof", or "lof" representing dominant-negative, gain-of-function, and loss-of-function mechanisms, respectively; pmid–the relevant PubMed identifier; evidence_line–a supporting statement for the mechanism in the published work.
(CSV)

**S2 Table. Model features, final hyperparameters, and common threshold metrics.** A collection of XLSX sheets containing information on the features used in the models, the final hyperparameters of the models, and for all models the estimates of common threshold metrics for every 0.01 increase in class probability.
(XLSX)

**S3 Table. Predicted molecular mechanisms for the human proteome.** The first sheet of this XLSX file contains column descriptions for the second sheet. The second sheet contains the molecular mechanism predictions.
(XLSX)

## Acknowledgments

We thank Benjamin J. Livesey for generating the VEP scores and providing helpful comments on the manuscript.

## Author Contributions

**Conceptualization:** Joseph A. Marsh.

**Data curation:** Mihaly Badonyi, Joseph A. Marsh.

**Formal analysis:** Mihaly Badonyi.

**Funding acquisition:** Joseph A. Marsh.

**Investigation:** Mihaly Badonyi, Joseph A. Marsh.

**Methodology:** Mihaly Badonyi, Joseph A. Marsh.

**Project administration:** Joseph A. Marsh.

**Resources:** Mihaly Badonyi, Joseph A. Marsh.

**Software:** Mihaly Badonyi, Joseph A. Marsh.

**Supervision:** Mihaly Badonyi, Joseph A. Marsh.

**Validation:** Mihaly Badonyi, Joseph A. Marsh.

**Visualization:** Mihaly Badonyi, Joseph A. Marsh.

**Writing – original draft:** Mihaly Badonyi.

**Writing – review & editing:** Mihaly Badonyi, Joseph A. Marsh.

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
