## [Decision Letter · Decision Letter 0]

2 May 2024

PONE-D-24-09287Proteome-scale prediction of molecular mechanisms underlying dominant genetic diseasesPLOS ONE

Dear Dr. Badonyi,

Thank you for submitting your manuscript to PLOS ONE. After careful consideration, we feel that it has merit but does not fully meet PLOS ONE’s publication criteria as it currently stands. Therefore, we invite you to submit a revised version of the manuscript that addresses the points raised during the review process.

We look forward to receiving your revised manuscript.

Kind regards,

Muhammad Farooq

Academic Editor

PLOS ONE

“European Research Council (ERC) under the European Union's Horizon 2020 research and innovation programme (grant agreement No. 101001169)

Biotechnology and Biological Sciences Research Council EASTBIO Doctoral Training Programme (BB/M010996/1).”

4. We notice that your supplementary figures are included in the manuscript file. Please remove them and upload them with the file type 'Supporting Information'. Please ensure that each Supporting Information file has a legend listed in the manuscript after the references list.

Additional Editor Comments:

Dear Prof. Mihaly Badonyi,

Thank you very much for your submission to Plos One. Your manuscript has been evaluated by two reviewers and the editors. Overall the comments are very positive regarding your manuscript but not acceptable in its present form, we would be willing to consider a revised manuscript, if you can address the comments of the reviewers. Please note that we are making no commitment to publish your manuscript before the revised manuscript review process.

With your response to comments, please provide each comment followed by your response. If substantive changes have been made to the manuscript, please provide a clear description of what you did and where. If you insert important sentences, paragraphs or sections in response to the comments, please include them in this response.

Reviewers' comments:

Reviewer's Responses to Questions

**Comments to the Author**

1. Is the manuscript technically sound, and do the data support the conclusions?

Reviewer #1: Yes

Reviewer #2: Yes

2. Has the statistical analysis been performed appropriately and rigorously? 

Reviewer #1: Yes

Reviewer #2: Yes

3. Have the authors made all data underlying the findings in their manuscript fully available?

Reviewer #1: Yes

Reviewer #2: Yes

4. Is the manuscript presented in an intelligible fashion and written in standard English?

Reviewer #1: Yes

Reviewer #2: Yes

5. Review Comments to the Author

Reviewer #1: The study entitled “Proteome-scale prediction of molecular mechanisms underlying dominant genetic diseases” is a reflecting good comprehensive work. However, improvement in language and clarity in sentences are needed. This work is acceptable with minor correction.

Comments:

As per OMIM, there are thousands of “dominant genetic diseases” grouped into autosomal and sex chromosome. Is this article addressing all as one category? What about recessive genetic diseases?

Author mentioned dominant-negative but ignored dominant-positive mutations.

Please enlist the interpretable features were used in the modeling study.

Reviewer #2: The manuscript entitled "Proteome-scale prediction of molecular mechanisms underlying Dominant Genetic Diseases" presents a comprehensive approach to predicting the molecular mechanisms underlying dominant genetic disorders. In this manuscript, the authors introduce a tripartite statistical model consisting of support vector machine binary classifiers trained to predict whether human protein-coding genes are associated with dominant-negative (DN), gain-of-function (GOF), or loss-of-function (LOF) molecular disease mechanisms. The study estimates the utility of these predictions by examining biologically and clinically important properties known to be associated with the mechanisms. The manuscript is well-written and provides valuable insights into predicting disease mechanisms at a proteome scale. The manuscript is acceptable for publication after minor revision.

Minor Comments

1. In introduction section It would be better to include a brief overview of existing computational approaches for predicting disease mechanisms and how the proposed tripartite statistical model improves upon or complements these methods.

2. Please include method layout diagram in the methodology section

3. Please improve the image quality of Figure 3

4. The inclusion of future directions and potential areas for further research would enrich the manuscript.

6. PLOS authors have the option to publish the peer review history of their article (what does this mean?). If published, this will include your full peer review and any attached files.

Reviewer #1: **Yes: **Sajjad Karim

Reviewer #2: **Yes: **Usman Ali Ashfaq

---

## [Author Response · Author response to Decision Letter 0]

24 May 2024

Reviewer #1: The study entitled “Proteome-scale prediction of molecular mechanisms underlying dominant genetic diseases” is a reflecting good comprehensive work. However, improvement in language and clarity in sentences are needed. This work is acceptable with minor correction.

Comments:

As per OMIM, there are thousands of “dominant genetic diseases” grouped into autosomal and sex chromosome. Is this article addressing all as one category? What about recessive genetic diseases?

Authors’ response

The study necessarily addresses dominant genes as one category. Our goal was to predict the most likely molecular mechanism that would be expected from a gene once dominant disease-associated mutations are identified in them. To this end, we collected reported mechanisms strictly in autosomal dominant genes. These genes may be haploinsufficient (i.e. dominant loss-of-function) or the mutated allele can give rise to a dominant-negative or an assembly-mediated gain-of-function effect. The majority of recessive mutations have a loss-of-function effect, with only a few examples of other mechanisms, e.g., recessive gain-of-function known to date. We have added a sentence in the introduction to bring this information to the reader’s attention: “Nearly all recessive disorders are associated with LOF, with only a few rare cases of alternative mechanisms identified to date [1]”.

Author mentioned dominant-negative but ignored dominant-positive mutations.

Authors’ response

The reviewer is correct that we did not create a separate category for the dominant-positive effect; there are two reasons for this. First, many dominant-positive effects are likely to have been grouped together with gain-of-function mechanisms in our data, because they essentially represent the assembly-mediated gain-of-function effect [1]. Second, the experimental challenges in precisely characterising molecular mechanisms mean that dominant-positive effects are often not distinctly reported in the literature, preventing their classification as a separate category.

Please enlist the interpretable features were used in the modeling study.

Authors’ response

All gene-level features used to train the models, along with their detailed descriptions, are available on the Open Science Framework repository (URL: https://osf.io/9gx5n; see the “features” tab in table_S2.xlsx). Additionally, with the resubmission of this manuscript to PLOS One, we have uploaded all referenced tables to be included in the supplementary material. 

Reviewer #2: The manuscript entitled "Proteome-scale prediction of molecular mechanisms underlying Dominant Genetic Diseases" presents a comprehensive approach to predicting the molecular mechanisms underlying dominant genetic disorders. In this manuscript, the authors introduce a tripartite statistical model consisting of support vector machine binary classifiers trained to predict whether human protein-coding genes are associated with dominant-negative (DN), gain-of-function (GOF), or loss-of-function (LOF) molecular disease mechanisms. The study estimates the utility of these predictions by examining biologically and clinically important properties known to be associated with the mechanisms. The manuscript is well-written and provides valuable insights into predicting disease mechanisms at a proteome scale. The manuscript is acceptable for publication after minor revision.

Minor Comments

1. In introduction section It would be better to include a brief overview of existing computational approaches for predicting disease mechanisms and how the proposed tripartite statistical model improves upon or complements these methods.

Authors’ response

To our knowledge, no gene-level molecular mechanism predictor has been developed thus far. In the original manuscript, under the Proteome-scale molecular mechanism prediction section, we included a paragraph and a supplemental figure (Figure S1) comparing our model to haploinsufficiency predictors. Our model is likely to be more specific than these predictors, as its component models were trained solely on dominant disease genes to tell apart alternative molecular mechanisms from that of LOF and vice versa.

To highlight this distinction and complement the analysis in the main text, we have added the following sentence to the Introduction: “Given the limitations of current methods that primarily focus on LOF mechanisms, such as haploinsufficiency predictors [49–51], we aimed to develop a method for more directly differentiating between dominant molecular mechanisms.”

We hope this revision addresses the reviewer’s concern.

2. Please include method layout diagram in the methodology section.

Authors’ response 

We appreciate the reviewer’s suggestion. We have added a workflow diagram (Figure 1 in the new manuscript) that illustrates the methodology and model validation.

3. Please improve the image quality of Figure 3.

Authors’ response 

We have now submitted TIF files in full resolution to the journal.

4. The inclusion of future directions and potential areas for further research would enrich the manuscript.

Authors’ response 

We have added a new paragraph to the Conclusion section to incorporate future directions for molecular disease mechanism prediction.

References (numbering follows the manuscript’s)

1. Backwell L, Marsh JA. Diverse Molecular Mechanisms Underlying Pathogenic Protein Mutations: Beyond the Loss-of-Function Paradigm. Annual review of genomics and human genetics. 2022;23. doi:10.1146/ANNUREV-GENOM-111221-103208

49. Izumi K, Nakato R, Zhang Z, Edmondson AC, Noon S, Dulik MC, et al. Germline gain-of-function mutations in AFF4 cause a developmental syndrome functionally linking the super elongation complex and cohesin. Nat Genet. 2015;47: 338–344. doi:10.1038/ng.3229

50. Seidman JG, Seidman C. Transcription factor haploinsufficiency: when half a loaf is not enough. J Clin Invest. 2002;109: 451–455. doi:10.1172/JCI15043

51. Kim J, Jiang H, Park S, Eltorai AEM, Stewart FR, Yoon H, et al. Haploinsufficiency of Human APOE Reduces Amyloid Deposition in a Mouse Model of Amyloid-β Amyloidosis. J Neurosci. 2011;31: 18007–18012. doi:10.1523/JNEUROSCI.3773-11.2011

---

## [Decision Letter · Decision Letter 1]

26 Jun 2024

Proteome-scale prediction of molecular mechanisms underlying dominant genetic diseases

PONE-D-24-09287R1

Dear Dr. Mihaly Badonyi,

We’re pleased to inform you that your manuscript has been judged scientifically suitable for publication and will be formally accepted for publication once it meets all outstanding technical requirements.

Kind regards,

Muhammad Farooq

Academic Editor

PLOS ONE

Additional Editor Comments (optional):

None

Reviewers' comments:

Reviewer's Responses to Questions

**Comments to the Author**

1. If the authors have adequately addressed your comments raised in a previous round of review and you feel that this manuscript is now acceptable for publication, you may indicate that here to bypass the “Comments to the Author” section, enter your conflict of interest statement in the “Confidential to Editor” section, and submit your "Accept" recommendation.

Reviewer #1: All comments have been addressed

Reviewer #2: All comments have been addressed

2. Is the manuscript technically sound, and do the data support the conclusions?

Reviewer #1: Yes

Reviewer #2: Yes

3. Has the statistical analysis been performed appropriately and rigorously? 

Reviewer #1: Yes

Reviewer #2: Yes

4. Have the authors made all data underlying the findings in their manuscript fully available?

Reviewer #1: Yes

Reviewer #2: Yes

5. Is the manuscript presented in an intelligible fashion and written in standard English?

Reviewer #1: Yes

Reviewer #2: Yes

6. Review Comments to the Author

Reviewer #1: (No Response)

Reviewer #2: The author has incorporated all suggested changes, addressing all previous comments comprehensively. The revised manuscript demonstrates significant improvement and is now acceptable for publication

7. PLOS authors have the option to publish the peer review history of their article (what does this mean?). If published, this will include your full peer review and any attached files.

Reviewer #1: **Yes: **Sajjad Karim

Reviewer #2: No

---

## [Editor Report · Acceptance letter]

5 Jul 2024

PONE-D-24-09287R1 

PLOS ONE

Dear Dr. Badonyi, 

I'm pleased to inform you that your manuscript has been deemed suitable for publication in PLOS ONE. Congratulations! Your manuscript is now being handed over to our production team.

Kind regards, 

on behalf of

Dr. Muhammad Farooq 

Academic Editor

PLOS ONE